# A Common Variant in the *CDK8* Gene Is Associated with Sporadic Pituitary Adenomas in the Portuguese Population: A Case-Control Study

**DOI:** 10.3390/ijms231911749

**Published:** 2022-10-04

**Authors:** Leonor M. Gaspar, Catarina I. Gonçalves, Fernando Fonseca, Davide Carvalho, Luísa Cortez, Ana Palha, Inês F. Barros, Ema Nobre, João S. Duarte, Cláudia Amaral, Maria J. Bugalho, Olinda Marques, Bernardo D. Pereira, Manuel C. Lemos

**Affiliations:** 1CICS-UBI, Health Sciences Research Centre, University of Beira Interior, 6200-506 Covilhã, Portugal; 2Serviço de Endocrinologia, Hospital de Curry Cabral, Centro Hospitalar Universitário Lisboa Central, 1050-099 Lisboa, Portugal; 3Serviço de Endocrinologia, Diabetes e Metabolismo, Centro Hospitalar de São João, Faculdade de Medicina e Instituto de Investigação e Inovação em Saúde da Universidade do Porto, 4200-319 Porto, Portugal; 4Serviço de Endocrinologia, Hospital de Braga, 4710-243 Braga, Portugal; 5Serviço de Endocrinologia, Diabetes e Metabolismo, Hospital de Santa Maria, Centro Hospitalar Universitário Lisboa Norte, 1649-028 Lisboa, Portugal; 6Serviço de Endocrinologia, Hospital de Egas Moniz, Centro Hospitalar Lisboa Ocidental, 1349-019 Lisboa, Portugal; 7Serviço de Endocrinologia, Hospital de Santo António, Centro Hospitalar Universitário do Porto, 4099-001 Porto, Portugal; 8Serviço de Endocrinologia e Nutrição, Hospital do Divino Espírito Santo de Ponta Delgada, 9500-782 Ponta Delgada, Portugal

**Keywords:** pituitary adenoma, NEBL, nebulette, PCDH15, protocadherin-related 15, CDK8, cyclin dependent kinase 8, SNP, single nucleotide polymorphism, genetic susceptibility

## Abstract

The majority of pituitary adenomas occur in a sporadic context, and in the absence of known genetic predisposition. Three common variants at the *NEBL* (rs2359536), *PCDH15* (rs10763170) and *CDK8* (rs17083838) loci were previously associated with sporadic pituitary adenomas in the Han Chinese population, but these findings have not yet been replicated in any other population. The aim of this case-control study was to assess if these variants are associated with susceptibility to sporadic pituitary adenomas in the Portuguese population. Genotype and allele frequencies were determined in 570 cases and in 546 controls. The *CDK8* rs17083838 minor allele (A allele) was significantly associated with sporadic pituitary adenomas, under an additive (odds ratio (OR) 1.73, 95% confidence interval (CI) 1.19–2.50, *p* = 0.004) and dominant (OR 1.82, 95% CI 1.24–2.68, *p* = 0.002) inheritance model. The *NEBL* rs2359536 and *PCDH15* rs10763170 variants were not associated with the overall risk for the disease, although a borderline significant association was observed between the *PCDH15* rs10763170 minor allele (T allele) and somatotrophinomas (dominant model, OR 1.55, 95% CI 1.02–2.35, *p* = 0.035). These findings suggest that the *CDK8* rs17083838 variant, and possibly the *PCDH15* rs10763170 variant, may increase susceptibility to sporadic pituitary adenomas in the Portuguese population.

## 1. Introduction

Pituitary adenomas, also referred to as pituitary neuroendocrine tumors (PitNETs), are the most common intracranial neoplasias [1]. Although usually benign, pituitary adenomas are associated with increased morbidity and mortality via hormone overproduction and mass effects resulting from compression of structures adjacent to the tumor [2]. Depending on the hormone expressed by the tumor cells, pituitary adenomas can be divided into non-functioning adenomas, prolactinomas, somatotrophinomas, corticotrophinomas, gonadotrophinomas and thyrotrophinomas [2].

The mechanisms underlying pituitary tumorigenesis are still largely unknown. A small proportion of tumors (~5%) are due to germline mutations, as part of syndromic diseases or as familial isolated pituitary adenomas [3,4]. These tumors are usually more aggressive, may present at a younger age, have a larger tumor size, show increased invasiveness, and are often resistant to standard treatments [5]. However, the vast majority of pituitary adenomas (~95%) occur in a sporadic context, in the absence of known genetic predisposition, and are likely to occur due to acquired somatic and epigenetic mutations [6].

A genome-wide association study (GWAS) in the Han Chinese population [7] identified single nucleotide polymorphisms (SNPs) at three independent loci (rs2359536 at 10p12.31, rs10763170 at 10q21.1 and rs17083838 at 13q12.13) that were significantly associated with sporadic pituitary adenomas. The genes around these susceptibility loci, namely the nebulette (*NEBL*), protocadherin-related 15 (*PCDH15*) and cyclin dependent kinase 8 (*CDK8*), are involved in cell–cell adhesion and regulation of cell cycle progression [7]. However, it is still uncertain if and how these loci are involved in pituitary tumorigenesis. Although the association with these variants was highly significant in the Han Chinese population, these results have not yet been confirmed in any other population.

The aim of this study was to investigate whether the *NEBL* rs2359536, *PCDH15* rs10763170 and *CDK8* rs17083838 SNPs are associated with the susceptibility to sporadic pituitary adenomas in the Portuguese population.

## 2. Results

The *NEBL* rs2359536, *PCDH15* rs10763170 and *CDK8* rs17083838 genotype and allele frequencies observed in the cases and controls are presented in Table 1. The observed genotype frequencies did not deviate from the Hardy–Weinberg equilibrium (data not shown), which would otherwise suggest selection bias, population stratification or genotyping errors. The *CDK8* rs17083838 minor allele (A allele) was significantly associated with overall sporadic pituitary adenomas under an additive (odds ratio (OR) 1.73, 95% confidence interval (CI) 1.19–2.50, *p* = 0.004) and dominant (OR 1.82, 95% CI 1.24–2.68, *p* = 0.002) inheritance model (Table 1). These associations remained statistically significant, even after applying the conservative Bonferroni correction (*p* < 0.0167). For the *NEBL* rs2359536 and *PCDH15* rs10763170 SNPs, no significant differences between cases and controls were observed.

In the analysis based on tumor subtype, the *PCDH15* rs10763170 minor allele (T allele) was associated with somatotrophinomas under a dominant inheritance model (OR 1.55, 95% CI 1.02–2.35, *p* = 0.035) (Table 2); however, this association was no longer significant after applying the Bonferroni correction. For the remaining tumor subtypes and sizes, no associations were observed (data not shown).

The combined number of *NEBL*, *PCDH15* and *CDK8* minor (risk) alleles that each individual harbored was not significantly associated with the risk of pituitary adenomas (Table 3).

## 3. Discussion

This case-control study revealed an increased frequency of the *CDK8* rs17083838 minor allele (A allele) in patients diagnosed with sporadic pituitary adenomas. Individuals with genotypes containing at least one minor (A) allele were associated with 1.82 times greater risk for sporadic pituitary adenomas, compared to individuals with the homozygous GG genotype (dominant inheritance model, OR 1.82, 95% CI 1.24–2.68, *p* = 0.002). These results suggest that this genetic variant, which was identified as a risk locus in a previous GWAS [7], is also a susceptibility variant for sporadic pituitary adenomas in the Portuguese population.

To date, there are no functional studies of the rs17083838 polymorphism, so it remains unclear by which mechanism this polymorphism influences the pathogenesis of pituitary adenomas. The rs17083838 polymorphism is located in the first intron of the *CDK8* gene, which is overexpressed in several cancers [8]. This gene encodes cyclin-dependent kinase 8 (CDK8), which belongs to the cyclin-dependent protein family that includes important regulators involved in cell cycle progression [8]. The CDK8 protein interacts with E2F1, protecting β-catenin from the inhibitory effect of E2F1, enabling its interaction with Wnt proteins [9]. The Wnt/β-catenin signaling pathway is important during development, regulates several aspects of cell proliferation, differentiation, and cell survival [10], and has been shown to be up-regulated in pituitary adenomas [11,12]. Furthermore, studies of animal models of pituitary tumors have shown that several target genes of CDK8 are differentially expressed in these tumors, suggesting a role of this kinase in pituitary tumorigenesis [13].

The *NEBL* rs2359536 (T > C) and *PCDH15* rs10763170 (C > T) variants were not found to be associated with overall sporadic pituitary adenomas in our study, despite their reported association in the Han Chinese population [7]. However, the *PCDH15* rs10763170 minor allele (T allele) presented a near-significant association with somatotrophinomas (dominant inheritance model, OR 1.55, 95% CI 1.02–2.35, *p* = 0.035). The rs10763170 polymorphism is located upstream of the *PCDH15* gene that encodes protocadherin-related 15. This protein is a member of the cadherin superfamily that is involved in cell–cell adhesion, morphogenesis, cell recognition and signaling [14], and in the development and progression of many cancers [15]. *PCDH15* is best known for its role in hereditary hearing loss (Usher syndrome type 1F) [16]. However, reduced expression of *PCDH15* has been shown to enhance oligodendrocyte progenitor cell proliferation and progression of gliomas [17]. Most importantly, PCDH15 has been shown to interact with cadherin-related 23 (CDH23) [18], which has been implicated in sporadic and familial forms of pituitary tumors [19]. Further studies are needed to determine whether *PCDH15* is also involved in pituitary tumorigenesis.

The exact mechanisms through which these SNPs contribute to the tumorigenesis of pituitary adenomas are still unknown. These genetic variants are located in non-coding regions, and it is presently unknown if they have a direct effect on gene expression or if they are in linkage disequilibrium with other nearby functional variants. Additional studies will be needed, to determine the functional consequences of these SNPs.

Our results partially validate those obtained by the GWAS in the Han Chinese population [7]. It remains to be determined if the observed associations can also be replicated in other European and non-European populations, with different environmental exposures and genetic profiles. In addition, studies with larger sample sizes may help to clarify if the associations are stronger with specific tumor subtypes.

In conclusion, our data suggest that the *CDK8* rs17083838 variant, and possibly the *PCDH15* rs10763170 variant, may increase susceptibility to sporadic pituitary adenomas in the Portuguese population. These findings may contribute to a better understanding of the genetic etiology of sporadic pituitary adenomas.

## 4. Materials and Methods

### 4.1. Subjects

The study was designed as a retrospective case-control association study. Cases comprised 570 patients with sporadic pituitary adenomas (275 males and 295 females; mean age at diagnosis ± standard deviation (SD) = 43.5 ± 16.3 years), recruited consecutively at endocrinology outpatient clinics in Portugal. Tumor classification was based on histological examination or, in the case of prolactinomas, by clinical, hormonal and radiological examination. Tumor subtypes were somatotrophinoma (n = 161), prolactinoma (n = 154), non-functioning (n = 138), gonadotrophinoma (n = 40), corticotrophinoma (n = 35), thyrotrophinoma (n = 4), mixed (n = 28) and undetermined (n = 10). Tumor size was based on the largest diameter, and was classified as macroadenoma (≥1 cm, n = 483), microadenoma (<1 cm, n = 65) and undetermined (n = 22). The control group comprised 546 unrelated healthy blood donors (264 males and 282 females; mean age ± SD = 38.1 ± 12.1 years), recruited at blood donation centers, with no known clinical history of pituitary disease, and originating from the same geographical regions as the patients. All the subjects were Caucasian Portuguese. Written informed consent was obtained from all the subjects, and the study was approved by the Ethics Committee of the Faculty of Health Sciences, University of Beira Interior (Ref: CE-UBI-Pj-2018-027).

### 4.2. Genetic Studies

The *NEBL* rs2359536, *PCDH15* rs10763170 and *CDK8* rs17083838 were selected for this study, as they had been demonstrated to be among those variants most strongly associated with sporadic pituitary adenomas in a previous GWAS [7]. Minor allele frequencies (MAFs) reported in the Genome Aggregation Database (gnomAD) [20] for Non-Finnish Europeans were 0.337 (rs2359536 allele C), 0.409 (rs10763170 allele T) and 0.059 (rs17083838 allele A), respectively. Venous blood samples were collected from each subject, and genomic deoxyribonucleic acid (DNA) was extracted from peripheral blood leukocytes, using the previously described methods [21]. Genotyping of the rs2359536, rs10763170 and rs17083838 SNPs was performed by a SNP genotyping assay with commercially available TaqMan probes (Assay ID: C-1974242-10 for rs2359536; Assay ID: C-411660-10 for rs10763170 and Assay ID: C-34697393-10 for rs17083838; Thermo Fisher Scientific, Waltham, MA, USA), according to the manufacturer’s instructions. The genotyping methods were validated by DNA sequencing of representative samples for each genotype (STAB VIDA, Caparica, Portugal; and ABI 3730XL, Applied Biosystems; Thermo Fisher Scientific, Waltham, MA, USA).

### 4.3. Statistical Analysis

The genotype and allele frequencies in the cases and controls were compared by chi-square tests and logistic regression analysis, to obtain odds ratios (ORs), 95% confidence intervals (CIs) and two-tailed *p*-values, using the SNP association tool SNPStats [22]. The analysis was adjusted for sex and age, to account for any effect of these variables on genotype and allele frequencies. The Hardy–Weinberg equilibrium of cases and controls was assessed by comparing the observed and allele-based expected genotype frequencies using SNPStats [22]. The best model of inheritance for each SNP (dominant, recessive, codominant or additive) was selected using Akaike’s Information Criterion (AIC) [22]. Subgroup analysis was carried out to assess the effect of the three polymorphisms on tumor subtype (each subtype vs. all others) and size (<1 vs. ≥1 cm of diameter). To determine the cumulative effect of minor (risk) alleles, each individual was classified as having zero, one or two minor alleles for each of the three SNPs. The total number of minor alleles per individual (ranging from a minimum of zero to a maximum of six) was then compared between cases and controls using a two-tailed chi-square test. A Bonferroni correction for multiple comparisons was used to correct statistical significance, which was set at *p* < 0.0167 (*p* < 0.05, divided by the number of analyzed SNPs). Power calculation was carried out using the software Power and Sample Size Calculations (version 3.1.6, Vanderbilt University, Nashville, TN, USA) [23]. Assuming minor allele frequencies of 34%, 41% and 6% it was estimated that the study sample was sufficient to detect ORs of 1.279 (rs2359536), 1.270 (rs10763170) and 1.576 (rs17083838), respectively, under an additive model of inheritance, with an estimated power of 0.8 and a type 1 error probability of 0.05.

## Figures and Tables

**Table 1 ijms-23-11749-t001:** Distribution of *NEBL*, *PCDH15* and *CDK8* genotypes and alleles in sporadic pituitary adenomas and controls.

*NEBL* rs2359536		Cases, n (%)	Controls, n (%)	OR (95% CI)	*p* Value	Adjusted OR (95% CI) ^†^	Adjusted *p* Value ^†^
Genotypes	TT	225 (39.5)	213 (39.0)				
	TC	271 (47.5)	258 (47.3)				
	CC	74 (13.0)	75 (13.7)	0.94 (0.66–1.32) ^¶^	0.71	0.89 (0.63–1.27) ^¶^	0.53
Alleles	T	721 (63.2)	684 (62.6)				
	C	419 (36.8)	408 (37.4)	0.97 (0.82–1.16) ^§^	0.76	0.95 (0.80–1.13) ^§^	0.57
***PCDH15* rs10763170**		**Cases, n (%)**	**Controls, n (%)**	**OR (95% CI)**	***p* Value**	**Adjusted OR (95% CI) ^†^**	**Adjusted *p* Value ^†^**
Genotypes	CC	184 (32.3)	180 (33.0)				
	CT	276 (48.4)	247 (45.2)				
	TT	110 (19.3)	119 (21.8)	0.86 (0.64–1.15) ^¶^	0.30	0.86 (0.64–1.15) ^¶^	0.31
Alleles	C	644 (56.5)	607 (55.6)				
	T	496 (43.5)	485 (44.4)	0.97 (0.82–1.14) ^§^	0.67	0.96 (0.81–1.13) ^§^	0.62
***CDK8* rs17083838**		**Cases, n (%)**	**Controls, n (%)**	**OR (95% CI)**	***p* Value**	**Adjusted OR (95% CI) ^†^**	**Adjusted *p* Value ^†^**
Genotypes	GG	487 (85.4)	498 (91.2)				
	GA	82 (14.4)	46 (8.4)				
	AA	1 (0.2)	2 (0.4)	1.77 (1.21–2.58) ^Ŧ^	0.003	1.82 (1.24–2.68) ^Ŧ^	0.002
Alleles	G	1056 (92.6)	1042 (95.4)				
	A	84 (7.3)	50 (4.6)	1.67 (1.16–2.41) ^§^	0.005	1.73 (1.19–2.50) ^§^	0.004

n, number; OR, odds ratio; CI, confidence interval. ^†^ adjusted for sex and age; ^¶^ recessive model; ^Ŧ^ dominant model; ^§^ log-additive model.

**Table 2 ijms-23-11749-t002:** Distribution of *PCDH15* genotypes and alleles in somatotrophinomas and all other tumors.

*PCDH15* rs10763170		Somatotrophinomas, n (%)	All Other Tumors, n (%)	OR (95% CI)	*p* Value	Adjusted OR (95% CI) ^†^	Adjusted *p* Value ^†^
Genotypes	CC	41 (25.5)	138 (34.6)				
	CT	85 (52.8)	193 (48.4)				
	TT	35 (21.7)	68 (17.0)	1.55 (1.03–2.33) ^Ŧ^	0.036	1.55 (1.02–2.35) ^Ŧ^	0.035
Alleles	C	167 (51.9)	469 (58.8)				
	T	155 (48.1)	329 (41.2)	1.32 (1.02–1.72) ^§^	0.035	1.31 (1.00–1.71) ^§^	0.046

n, number; OR, odds ratio; CI, confidence interval. ^†^ adjusted for sex and age; ^Ŧ^ dominant model; ^§^ log-additive model.

**Table 3 ijms-23-11749-t003:** Distribution of combinations of minor (risk) alleles.

Number of Minor Alleles	Cases, n (%)	Controls, n (%)	OR (95% CI)	*p* Value
0	56 (9.8)	60 (11.0)		
1	180 (31.6)	176 (32.2)		
2	208 (36.5)	187 (34.2)		
3	102 (17.9)	102 (18.7)		
4	23 (4.0)	18 (3.3)		
5	1 (0.2)	3 (0.6)		
6	0 (0.0)	0 (0.0)		
2 or less	444 (77.9)	423 (77.5)		
3 or more	126 (22.1)	123 (22.5)	0.98 (0.74–1.29)	0.866

n, number; OR, odds ratio; CI, confidence interval.

## Data Availability

The data that support the findings of this study are available from the corresponding author on reasonable request.

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
