# Peer review of "A Common Variant in the CDK8 Gene Is Associated with Sporadic Pituitary Adenomas in the Portuguese Population: A Case-Control Study"

_ijms, 2022, doi:10.3390/ijms231911749_

Round 1
Reviewer 1 Report
An article “A common variant in the CDK8 gene is associated with sporadic pituitary adenomas in the Portuguese population: a case-control study” by Leonor M. Gaspar et al is a very interesting and novel study investigating genetic predisposition of sporadic pituitary tumors in Portuguese population.
Three common variants at the NEBL (rs2359536), PCDH15 29 (rs10763170) and CDK8 (rs17083838) loci previously associated with sporadic pituitary tumors in the Han Chinese population were studied. The results of the study suggest that the CDK8 rs17083838 variant, and possibly the PCDH15 rs10763170 variant, may increase the susceptibility to sporadic pituitary adenomas in the Portuguese population.
The study was designed as a retrospective case-control association study, with a substantial number of patients recruited. The manuscript is very well written.
However, in regard to the study design very important basic information ( e.g. age, sex, ethnicity) on both, the patients with sporadic tumors and patients from control group is missing. Assumption is that control group patients (blood donors) have no history of pituitary tumors, however this information about control group is missing.
In regard to the study design and specificity detail information about the methods of recruitment and studied population should be completed.
Reviewer 2 Report
The authors found that CDK8 SNP is associated with sporadic pituitary adenomas.
This result adds to the results in the Chinese population and confirms part of them. This is an interesting result, but we wish they would investigate more about the pathophysiology or mechanisms underlying the disease. Also, analysis of larger populations would be necessary to draw more definitive conclusions about those mutations, and their additive effect.
